# High-Light versus Low-Light: Effects on Paired Photosystem II Supercomplex Structural Rearrangement in Pea Plants

**DOI:** 10.3390/ijms21228643

**Published:** 2020-11-16

**Authors:** Alessandro Grinzato, Pascal Albanese, Roberto Marotta, Paolo Swuec, Guido Saracco, Martino Bolognesi, Giuseppe Zanotti, Cristina Pagliano

**Affiliations:** 1Department of Biomedical Sciences, University of Padova, Via Ugo Bassi 58 B, 35121 Padova, Italy; alessandro.grinzato@studenti.unipd.it (A.G.); giuseppe.zanotti@unipd.it (G.Z.); 2Applied Science and Technology Department–BioSolar Lab, Politecnico di Torino, Environment Park, Via Livorno 60, 10144 Torino, Italy; pascal.albanese@gmail.com (P.A.); guido.saracco@polito.it (G.S.); 3Center for Convergent Technologies, Electron Microscopy Facility, Istituto Italiano di Tecnologia—IIT, Via Morego 30, 16163 Genova, Italy; roberto.marotta@iit.it; 4Department of BioSciences, University of Milano, Via Celoria 26, 20133 Milano, Italy; paolo.swuec@fht.org (P.S.); Martino.Bolognesi@unimi.it (M.B.); 5Cryo-Electron Microscopy Facility, Human Technopole, Via Cristina Belgioioso 171, 20157 Milano, Italy

**Keywords:** cryo-electron microscopy, light acclimation, photosystem II supercomplex, plant thylakoid membranes

## Abstract

In plant *grana* thylakoid membranes Photosystem II (PSII) associates with a variable number of antenna proteins (LHCII) to form different types of supercomplexes (PSII-LHCII), whose organization is dynamically adjusted in response to light cues, with the C_2_S_2_ more abundant in high-light and the C_2_S_2_M_2_ in low-light. Paired PSII-LHCII supercomplexes interacting at their stromal surface from adjacent thylakoid membranes were previously suggested to mediate *grana* stacking. Here, we present the cryo-electron microscopy maps of paired C_2_S_2_ and C_2_S_2_M_2_ supercomplexes isolated from pea plants grown in high-light and low-light, respectively. These maps show a different rotational offset between the two supercomplexes in the pair, responsible for modifying their reciprocal interaction and energetic connectivity. This evidence reveals a different way by which paired PSII-LHCII supercomplexes can mediate *grana* stacking at diverse irradiances. Electrostatic stromal interactions between LHCII trimers almost completely overlapping in the paired C_2_S_2_ can be the main determinant by which PSII-LHCII supercomplexes mediate *grana* stacking in plants grown in high-light, whereas the mutual interaction of stromal N-terminal loops of two facing Lhcb4 subunits in the paired C_2_S_2_M_2_ can fulfil this task in plants grown in low-light. The high-light induced accumulation of the Lhcb4.3 protein in PSII-LHCII supercomplexes has been previously reported. Our cryo-electron microscopy map at 3.8 Å resolution of the C_2_S_2_ supercomplex isolated from plants grown in high-light suggests the presence of the Lhcb4.3 protein revealing peculiar structural features of this high-light-specific antenna important for photoprotection.

## 1. Introduction

Plants experience a highly fluctuating light environment, with irradiances varying seasonally, diurnally and spatially, at the level either of the plant canopy or the leaf. Variations in light intensity lead to considerable changes in leaf morphology and structure (e.g., modification of leaf mass/area ratio, development of sun-type or shade-type chloroplasts) [1], accompanied by dynamic modifications of leaf orientation [2] and chloroplasts position in the cell [3]. At chloroplast level, long-term high-light acclimated (or sun) plants with respect to low-light acclimated (or shade) plants show an increased number of chloroplasts per cell with fewer thylakoids per *granum* and narrower *grana* stacks [1,4]. At a molecular scale, Photosystem II (PSII), the water-splitting enzyme of photosynthesis [5], among all the protein complexes embedded in the thylakoid membranes is the major target of light acclimation [6]. In plants, PSII is mainly located in the *grana* domain of the thylakoid membranes [7] where it associates, as a dimeric core complex (C_2_), with a variable number of light-harvesting complex II (LHCII) trimers. The latter bind strongly (i.e., S-trimers) or moderately (i.e., M-trimers) to form different types of PSII-LHCII supercomplexes (PSII-LHCIIsc) [8,9]. In the C_2_S_2_ supercomplex the PSII dimeric core binds two LHCII S-trimers via two copies of the monomeric Lhcb4 and Lhcb5 subunits [8]. Larger PSII-LHCIIsc, known as C_2_S_2_M and C_2_S_2_M_2_, contain, respectively, one or two additional LHCII M-trimer(s) bound via the monomeric Lhcb4 and Lhcb6 subunits [8]. The amount and composition of the PSII-LHCIIsc in the thylakoid membranes are dynamically adjusted in response to changes in light intensity, with the C_2_S_2_ more abundant in high-light, whereas the C_2_S_2_M and C_2_S_2_M_2_ more represented at moderate and low-light, respectively [10,11,12]. In this regard, a reduction of Lhcb3, Lhcb6 and M-LHCII trimers bound to the PSII cores, in concomitance with an increase of Lhcb4.3 (also renamed Lhcb8 [13,14]), a specific high-light induced isoform of Lhcb4 [11,15,16,17], are the major players in the modulation of the PSII-LHCIIsc antenna size in plants long-term acclimated to increased light levels [10,11]. In the *grana* membranes, PSII-LHCIIsc are mostly (>90%) uniformly randomly distributed, with a higher density in high-light acclimated plants with respect to low-light acclimated ones [10]. Notwithstanding, in certain conditions, the C_2_S_2_M_2_, C_2_S_2_M and C_2_S_2_ PSII-LHCIIsc have been reported to form within the membrane well-ordered semi-crystalline arrays made of the same type of supercomplex, finally arranging in several different types of packing (see review [18]). With regards to the light, these semi-crystalline arrays account for up to about 10–20% of the PSII-LHCIIsc particles in the *grana* of plants grown in low-light and conversely show a markedly lower frequency under prolonged acclimation to increasing light intensities [10,19].

Recently, the high-resolution structures of plant C_2_S_2_ [20] and C_2_S_2_M_2_ [21,22] obtained by cryo-electron microscopy (cryo-EM) provided the arrangement of all proteins and pigments, necessary to construct models of energy transfer in the PSII-LHCIIsc [23]. At the next level of organization, PSII-LHCIIsc of type C_2_S_2_M_2_ [22], C_2_S_2_M [24] and C_2_S_2_ [25], isolated directly from plant thylakoid membranes under physiologically-relevant cation concentrations preserving the stacked morphology of the *grana* [26], were found to form pairs interacting across their stromal surface. In the case of the paired C_2_S_2_M supercomplex, this interaction was proven to occur through a specific overlap between apposing LHCII trimers interacting with their N-terminal loops and via physical connections spanning the stromal gap made by Lhcb4 subunits facing each other in adjacent membranes [17,24]. Contacts between PSII-LHCIIsc located in adjacent thylakoid membranes have been detected also in situ by cryo-electron tomography (cryo-ET) in semi-crystalline arrays of C_2_S_2_ supercomplexes occasionally occurring in plants high-light illuminated [7]. Notably, despite the different geometry of interaction displayed by the C_2_S_2_ supercomplexes in these semi-crystalline arrays occurring in local patches of membrane [7] with respect to the particles isolated from the whole stacked thylakoids [25], a similar overlapping of the LHCII trimers was observed, supporting the hypothesis that the electrostatic stromal interactions of LHCII trimers play a key role in *grana* stacking [27,28]. *Grana* are a characteristic feature of vascular plant chloroplasts [29] and ubiquitous in any irradiance [1]. The stacking of *grana* is thought to be driven by adhesion of LHCII trimers in adjacent membranes [29] and by contacts of PSII-LHCIIsc in adjacent thylakoids through the stromal surfaces of both LHCIIs and PSII cores [7,17]. The involvement of paired C_2_S_2_M supercomplexes in maintaining the *grana* stacking has been recently proven in pea plants grown in moderate light [17,24]. For a comprehensive understanding of the modulation of the *grana* stacking in relation to the different PSII-LHCIIsc organization at changing irradiances, we extended the structural study to paired supercomplexes of type C_2_S_2_ and C_2_S_2_M_2_ isolated from pea plants grown respectively at high and low irradiances. We base this structural study on our previous in-depth biochemical characterization of both the isolated PSII-LHCIIsc [11] and their sourcing stacked thylakoid membranes [16] extracted from pea plants grown in the same high-light and low-light conditions, where the C_2_S_2_ and the C_2_S_2_M_2_ are respectively the most abundant supercomplexes. We rely also on the previous detailed structural descriptions of the paired C_2_S_2_M supercomplex isolated from pea plants grown in moderate irradiance [24] and the specific stromal interactions occurring either on its S-trimer side or M-trimer side [17]. Here, by using single particle cryo-EM, we show the effects of the light-induced structural rearrangement of the paired PSII-LHCIIsc deduced by the maps of the paired C_2_S_2_ and C_2_S_2_M_2_ isolated from plants grown respectively at high and low irradiance, and obtained respectively at 6.5 Å and 13.1 Å resolution. The light-induced structural changes in the paired PSII-LHCIIsc modify the stromal interactions between the two moieties of the supercomplex and their energetic connectivity, possibly enabling the maintenance of *grana* stacking at diverse irradiances. By applying cryo-ET to the thylakoid membranes used to purify the PSII-LHCIIsc, we further validate the occurrence of paired PSII-LHCIIsc facing each other across the stromal gap in the isolated thylakoid membranes that largely retain their stacked organization. In addition, we show that the stromal distance measured in the detergent-solubilized paired PSII-LHCIIsc is compatible with the stromal distance observed between two adjacent thylakoid membranes, thus making feasible their involvement in *grana* stacking through interaction between flexible N-terminal loops of facing LHCIIs. Moreover, by reconstructing one of the two moieties of the paired C_2_S_2_ supercomplex isolated from plants grown in high-light we solved its structure at 3.8 Å resolution, suggesting the presence of the Lhcb4.3 protein and revealing peculiar structural features of this high-light-specific Lhcb4 isoform important for photoprotection.

## 2. Results and Discussion

### 2.1. Cryo-EM Maps of Paired PSII-LHCII Supercomplexes Isolated from Plants Grown under High-Light and Low-Light

The differentiation of the thylakoid membranes into *grana* and stroma lamellae is a prominent and ubiquitous feature of vascular plant chloroplasts [30]. Depending on the light condition, the ratio of granal to stromal membranes and the number and diameter of discs within *grana* stacks are highly variable. Sun and high-light grown plants with respect to shade and low-light grown plants have more *grana* per chloroplast with fewer discs per *granum* (5–16 versus up to 160) with smaller *granum* diameter (ranging between 300–600 nm) [1,31].

Previously, we showed how paired PSII-LHCIIsc of type C_2_S_2_M interact with each other within *grana* stacks [24]. For that study, as starting material we used stacked thylakoids isolated from pea plants grown at moderate light intensity (i.e., 150 µmol photons m^−2^ s^−1^), where the C_2_S_2_M is the most abundant supercomplex [11]. In this study, to investigate the effect of light intensity on the arrangement of paired PSII-LHCIIsc of different type and their stromal interaction within *grana* stacks, we purified paired PSII-LHCIIsc from pea plants grown in high-light (i.e., 750 µmol photons m^−2^ s^−1^, H) and low-light (i.e., 30 µmol photons m^−2^ s^−1^, L), where the C_2_S_2_ and the C_2_S_2_M_2_ are, respectively, the most abundant supercomplexes (see our previous dedicated paper, Figure 2 in Reference [11]). In all these preparations, thylakoid extraction was performed in the presence of divalent cation concentrations (i.e., 5 mM Mg^2+^) that resemble the native chloroplast ionic environment [32], to preserve the stacked morphology of the *grana* membranes. The subsequent purification of the paired PSII-LHCIIsc was performed by a quick (1 min) and direct solubilization of stacked thylakoids with the detergent *n*-dodecyl-α-D-maltoside (α-DDM) followed by sucrose density gradient ultracentrifugation in the presence of the same concentration of divalent cations, according to an optimized protocol previously reported [33] with slight modifications (see the Section 4 for details). According to this separation, a band corresponding to paired PSII-LHCIIsc made of a mixture of paired C_2_S_2_M_2_, C_2_S_2_M, and C_2_S_2_ has been obtained (i.e., band α3 in Appendix A), that is characterized by a majority of C_2_S_2_ in the high-light sample and C_2_S_2_M_2_ in the low-light sample as previously reported (for a detailed biochemical description of this preparation, and the starting thylakoid membranes, see our previous dedicated paper [11]). Subsequently, the detergent-solubilized paired PSII-LHCIIsc were frozen in vitreous ice and then imaged by transmission electron microscopy. A cryo-EM dataset was collected for each light condition (i.e., H and L datasets) and single particle analysis was performed (for technical details see Section 4 and Appendix A). The (C_2_S_2_)_2_ (classes 1 and 2, corresponding to 49% of the total particles, Appendix A) and the (C_2_S_2_M_2_)_2_ (class 1, corresponding to 40% of the total particles, Appendix A) resulted the most abundant supercomplexes in high-light and low-light, respectively, in accordance with the biochemical characterization of the two preparations previously reported (see Figure 2 in [11]). An initial 3D reconstruction of the paired C_2_S_2_ in high-light (hereafter (C_2_S_2_)_2_-H) was limited to 6.5 Å resolution (class 1 in Appendix A, Figure 1A). A small fraction of paired C_2_S_2_ particles allowed the reconstruction of a map at 8.4 Å resolution with visible stromal connections (class 2 in Appendix A, Figure 1B). Similarly, a map of the paired C_2_S_2_M_2_ in low-light (hereafter (C_2_S_2_M_2_)_2_-L), at final estimated resolution of 13.1 Å, showed two C_2_S_2_M_2_ interacting through physical connections spanning the stromal gap (class 1 in Appendix A, Figure 1B). These connections were also discernible in the 11 Å cryo-EM map of paired C_2_S_2_M in high-light (hereafter (C_2_S_2_M)_2_-H) (class 3 in Appendix A, Figure 1B), and closely resembled those in the moderate irradiance counterpart (EMD-3825) [24]. In all the maps, the stromal distance between the paired supercomplexes was ~ 20 Å and the stromal connections observed were always centrally-located and with a shape that seems related to the specific arrangement of the paired PSII-LHCIIsc.

### 2.2. In Situ Arrangement of Paired PSII-LHCII Supercomplexes in Isolated Stacked Thylakoid Membranes

To test if the thylakoid membranes used to purify the paired PSII-LHCIIsc retain a close-to-native organization, we performed cryo-ET on thylakoid membranes isolated from pea plants grown in high-light. Tomograms of vitrified isolated thylakoid membranes showed that, despite the occurrence of some swollen thylakoids especially at the *grana* ends, the membranes largely retained their stacked conformation, with a number of discs within the *grana* ranging between 3 and 10 (Figure 2A,D,H, Appendix A). The stromal gap and the thylakoid lumen were easily distinguishable, with the former noticeably darker than the latter, suggesting a higher protein content. In side view the *grana* stacks clearly showed characteristic densities associated with the thylakoid membrane, corresponding to the ATP-synthase protruding at the *grana* end membranes (Figure 2A,D,H white arrowheads) and to the extrinsic subunits of PSII protruding into the thylakoid lumen (Figure 2A,D,H black arrowheads). In tangential viewed *grana* stacks the PSII-LHCIIsc were clearly distinguishable, appearing closely packed and randomly distributed in the thylakoid membrane (white arrows in Figure 2D and Insets of Figure 2A,H). The crowding of PSII-LHCIIsc in the *grana* membranes was evident also in side view (Figure 2A,D,H). Here, when considering two adjacent thylakoid membranes, the PSII-LHCIIsc appeared either as two apposing particles forming pairs across the stromal gap or as single particles (Figure 2B,C,E–G). These results exemplify both the maintenance of stacking of the isolated thylakoids and the occurrence therein of paired PSII-LHCIIsc. The presence of both single and paired PSII-LHCIIsc within the thylakoid membranes is in agreement with the biochemical characterization of sucrose density gradient fractions of the α-DDM-treated sourcing thylakoids (Appendix A), showing the presence of two distinct bands containing the paired PSII-LHCIIsc and the unpaired/broken PSII-LHCIIsc (i.e., respectively band α3 and α2 in Appendix A; for a full description of this preparation see our previous dedicated paper [11]). Visual inspection of the tomograms revealed a mostly random distribution of PSII-LHCIIsc within the membrane, with an overall non-crystalline appearance (Figure 2D and insets of Figure 2A,H). Thus, the observed pairing of PSII-LHCIIsc in adjacent thylakoid membranes is independent from the formation of 2D arrays of PSII-LHCIIsc. However, this does not exclude that small portions of 2D arrays of PSII-LHCIIsc exist in isolated patches of the thylakoid membranes of pea plants grown in high-light and that higher quantity might exist in thylakoids of pea plants grown in low-light, as expected from literature [10,19].

To further test if the isolated thylakoid membranes preserved also their fine organization maintaining the correct spacing between adjacent *grana* membranes, we measured the distance between two adjacent membranes in more than 10 tomograms of *grana* stacks. Based on 500 measurements, the width of the stromal gap was 3.0 ± 0.5 nm and the center-to-center distance between adjacent thylakoid membranes across the stromal gap was 7.6 ± 0.7 nm (Figure 2I). These results are in agreement with similar measurements performed on tomograms of vitrified intact chloroplasts of pea plants (respectively 3.2 nm and 7.2 nm [7]), and with other published measurements of stromal gap (e.g., between 3.3–3.6 nm in *Arabidopsis thaliana* [34]; ~3 nm in *Chlamydomonas reinhardtii* [35]). Moreover, the width of the stromal gap measured in the thylakoid membranes from which the PSII-LHCIIsc have been purified was compatible with the stromal distance (~20 Å) between the two moieties of the detergent-solubilized paired PSII-LHCIIsc cryo-EM maps (see Figure 1). Indeed, considering an approximate membrane thickness of 4 nm and a height of 5 nm of the portion of the cryo-EM maps corresponding to the transmembrane components of each moiety of the paired PSII-LHCIIsc (Figure 1), the stromal gap distance of the detergent-solubilized PSII-LHCIIsc of ~2 nm is compatible with its counterpart of ~3 nm in the sourcing thylakoid membranes. Indeed, the portion of the cryo-EM maps corresponding to the transmembrane components of the paired PSII-LHCIIsc accounts also for the terminal alfa-helix portions of proteins protruding outside the membrane, clearly contributing to a partial occupancy of the stromal gap itself. Overall these tomographic results indicate that the stromal surfaces of *grana* thylakoids remained tightly appressed and substantially undisturbed by the procedure adopted for thylakoid isolation and that specific interactions can occur therein between the two facing PSII-LHCIIsc across the stromal gap. These results also suggest that paired PSII-LHCIIsc in non-crystalline *grana* thylakoids can be directly involved in *grana* stacking as well as the PSII-LHCIIsc present in the less common 2D arrays [7].

### 2.3. Peculiar Structural Features of the C_2_S_2_ Supercomplex in Plants Long-Term Acclimated to High-Light

The dynamic control of *grana* stacking is crucial for the photosynthetic adaptation of plants to light cues [36,37] and, under variable light intensities, depends on the reversible macro-reorganization of PSII-LHCIIsc [36]. In response to long-term exposure to different light intensities, the macro-reorganization of the PSII-LHCIIsc is determined by the modulation of its protein composition. In plants long-term acclimated to increased light levels a reduction of Lhcb3, Lhcb6 and M-LHCII trimers bound to the PSII cores, in concomitance with an increase of Lhcb4.3, are the major players in the modulation of the PSII-LHCIIsc macro-reorganization, that changes progressively from predominantly a C_2_S_2_M_2_ to a C_2_S_2_ [10,11]. Noteworthily, the Lhcb4.3 isoform, despite the close homology with the Lhcb4.1/Lhcb4.2, has a unique sequence and expression profile [13,14,38,39]. Although previously denoted as a “rarely expressed” LHC protein due to its low transcript level [13], the Lhcb4.3 isoform seems to have a specific role in long-term high-light acclimation, as attested by its increase either at transcript level [40,41] or at protein level [11,15,16,17] in plants exposed to high irradiance.

The accumulation of the Lhcb4.3 protein in preparations of PSII-LHCIIsc analogous to those used in this structural work, and isolated from pea plants grown in the same light conditions, has been previously determined by bottom-up mass spectrometry, revealing a 6-fold higher abundance of Lhcb4.3 in the high-light sample with respect to the low-light sample [11], with a predominant localization in the C_2_S_2_ supercomplex [42]. In addition, the accumulation of Lhcb4.3 in high-light was also confirmed in the sourcing thylakoid membranes used for isolation of these PSII-LHCIIsc [16]. In this work, to investigate the structural features of the C_2_S_2_ supercomplex in high-light, we decided to mask one of the two moieties of the (C_2_S_2_)_2_-H map obtained at 6.5 Å (class 1 in Appendix A, Figure 1A) and reconstruct a single C_2_S_2_. In this way, it was possible to obtain a 3.8 Å resolution map of the C_2_S_2_ supercomplex isolated from pea plants grown in high-light (hereafter C_2_S_2_-H) (Figure 1A). In our structural model, the significant reduction of the map density signal in the region of the C-terminus of the Lcb4 (Figure 3B) suggests the presence of the Lhcb4.3 isoform alongside the Lhcb4.1/Lhcb4.2 isoform. This structural observation is in agreement with biochemical evidences of accumulation of the Lhcb4.3 isoform in PSII-LHCIIsc predominantly of type C_2_S_2_ [11,42] and characterization of this isoform sequence [17] in analogous preparations obtained from pea plants grown in the same high-light conditions. Indeed, as recently shown by top-down mass spectrometry [17], the Lhcb4.3 sequence in *Pisum sativum* differs at the C-terminus for the lack of 14 amino acids and the occurrence of several substitutions with respect to the Lhcb4.1/Lhcb4.2 counterpart (Appendix A), which so far is the only Lhcb4 isoform resolved either in plant PSII-LHCIIsc [20,21,22] or as purified protein [43]. Considering the loss of density in the Lhcb4 C-terminus region of the C_2_S_2_-H map and the biochemical evidences of accumulation of the C-terminal truncated Lhcb4.3 isoform predominantly in the C_2_S_2_ in pea plants grown in high-light [11,17,42], due to the relatively low resolution of this map zone (5–6 Å), to trace the amino acids in the corresponding structural model we used the Lhcb4.3 amino acid sequence obtained by top-down mass spectrometry [17] (Figure 3 and Appendix A). Proteins in the PSII-LHCIIsc serve mainly for the fine positioning of pigment molecules, finally determining the excitation energy pathways from the outer LHCII antenna to the PSII core. Comparing the C_2_S_2_-H density map with the C_2_S_2_ [20] from spinach and the C_2_S_2_ module in the C_2_S_2_M_2_ [22] from pea plants grown under unspecified, likely moderate irradiances, it was possible to notice that some chlorophyll molecules showed less defined or absent density map, also taking in account the map quality of the neighborhood chlorophylls (Figure 3C). The identities of these chlorophylls were assigned according to their similarity in the corresponding sites in the pea high-resolution structure [22] obtained from samples purified in conditions very similar to ours [33] (Appendix A). The exclusive targets of this chlorophyll rearrangement were the monomeric Lhcb4 and Lhcb5 subunits (Figure 3D). Lhcb5 was slightly affected, with one less defined chlorophyll, putative Chl*b*_608_, on the distal part of the supercomplex at the interface with the S-trimer. Lhcb4 presented four units of less defined or missing chlorophyll: putative Chl*a*_616_ at the binding interface with the inner antenna protein CP47 of the PSII core, putative Chls *a*_601_ and *b*_614_ located at the binding interface with Lhcb6 and putative Chl*a*_611_ at the binding interface with the M-trimer. This difference in the density map can be explained by a reduction in the population of these chlorophylls or by an increase of their mobility (Appendix A). The structurally altered Lhcb4.3 interaction interface with Lhcb6 due to the truncated Lhcb4.3 C-terminus represents a peculiar feature of plants long-term exposed to high-light that likely serves to constitutively reduce the binding affinity for additional M-trimers under excessive irradiation, resulting in a predominant C_2_S_2_ organization of the PSII-LHCIIsc [42].

### 2.4. The Different Structural Arrangement of Paired PSII-LHCII Supercomplexes in High-Light and Low-Light Modifies Their Structural Interaction and Energetic Connectivity

Looking at the cryo-EM maps of paired PSII-LHCIIsc from plants grown in high-light and low-light, the most striking difference observed regards the relative orientation of each supercomplex within the pair (Figure 4). It is worth noting that this difference is macroscopic, despite the different intermediate resolutions of the maps. Comparing the models of the (C_2_S_2_)_2_ (Figure 4A) and (C_2_S_2_M_2_)_2_ (Figure 4C) structures for the high-light and low-light, respectively, a different rotational offset between the two supercomplexes around the membrane plane’s normal vector is evident. Specifically, this offset is ~44° in the (C_2_S_2_)_2_-H (Figure 4D) and ~2° in the (C_2_S_2_M_2_)_2_-L (Figure 4F). An intermediate offset of ~24° (Figure 4E) characterizes the (C_2_S_2_M)_2_, as visible in the map of the (C_2_S_2_M)_2_-H (Figure 4B) and similarly displayed by its moderate irradiance counterpart (EMD-3825) [24]. Considering the prevalent abundance of the (C_2_S_2_M_2_)_2_, (C_2_S_2_M)_2_ and (C_2_S_2_)_2_ in low, moderate, and high irradiances, respectively [11], a correspondence is evident between the systematic increase in the offset at increasing light intensity and the progressive loss of M-trimers in each supercomplex within the pair. In turn, this determines two main structural rearrangements of the paired PSII-LHCIIsc across the stromal gap. In the (C_2_S_2_)_2_-H, facing PSII dimeric cores show the highest angular divergence and an almost complete overlap of apposing S-trimers occurs (Figure 4A,D). This overlap resembles that occurring at the S-trimer side of the paired C_2_S_2_M supercomplex isolated from pea plants grown in moderate irradiance [24], where the N-terminal tail networks of facing Lhcb1 and Lhcb2 subunits are responsible for the stromal interactions of the LHCII trimers [17]. Based on these observations, it is likely that interactions of N-terminal loops of Lhcb1 and Lhcb2 subunits located in facing S-trimers may function as major determinant to lock in place the two apposing C_2_S_2_ supercomplexes in plants in high-light. In the (C_2_S_2_M_2_)_2_-L, facing PSII dimeric cores display the lowest angular divergence and facing Lhcb4 subunits from each supercomplex are in close proximity (Figure 4C,F). This proximity, which occurs in all the paired supercomplexes containing facing M-trimers (Figure 4B,C,E,F), allows the mutual interactions between the Lhcb4 long stroma-exposed N-terminal loops [17], which may be primarily responsible for anchoring together the paired PSII-LHCIIsc in plants in low-light. Considering that >80% of the *grana* membrane is occupied by protein complexes, most of which are fully assembled PSII-LHCIIsc [44], the specific modifications in the interactions between apposing supercomplexes with different arrangement here described likely represent an important mechanism by which paired PSII-LHCIIsc maintain the *grana* stacking at different irradiances. In high-light acclimated plants, characterized by fewer discs per *granum* [1,31] and higher density of PSII-LHCIIsc in the membrane [10] with respect to low-light acclimated plants, the almost complete overlapping of S-trimers driven by electrostatic interactions (i.e., the so-called “Velcro effect” [27,28]) in the prevalent (C_2_S_2_)_2_ supercomplex may be determinant for the *grana* stacking maintenance. In low-light acclimated plants, characterized by a higher number of discs per *granum* [1,31] and lower density of the of PSII-LHCIIsc in the membrane [10], this task can be fulfilled by the stromal connections made by N-term loops of facing Lhcb4 in the prevalent (C_2_S_2_M_2_)_2_ supercomplexes, by acting as physical “anchors” between two adjacent thylakoid membranes.

The excitation energy transfer between chromophores is a function of the distance between the pigments [45]. To investigate the effect of the different reciprocal arrangement of PSII-LHCIIsc within the pair (Figure 4) on the possible energy transfer between stromal chlorophylls of adjacent supercomplexes, we measured the Mg-Mg distance between any possible Chls pair within a maximum inter-pigment separation threshold of 90 Å (Appendix A), the distance at which the efficiency of the excitation transfer is 50% [45]. From this analysis, an overall increase in the number of these Chls pairs was observed in paired PSII-LHCIIsc with bigger antenna (Figure 5A). Moreover, considering the transition from the (C_2_S_2_M_2_)_2_ to the (C_2_S_2_)_2_, a progressive reduction in the average distance between the Chls pairs of the LHCIIs (Figure 5B), due to the major overlap of apposing S-S trimers in the (C_2_S_2_)_2_ with respect to the S-M trimers in the larger supercomplexes (Figure 4), was counterbalanced by a progressive increase in the distances of the Chls pairs of the PSII cores (Figure 5B), according to the minor overlap of facing PSII cores in the smaller supercomplexes (Figure 4). These evidences suggest a facilitated transversal energy transfer in the (C_2_S_2_M_2_)_2_ with respect to (C_2_S_2_)_2_ supercomplexes, which in turn might reflect an increased PSII connectivity. Analyses of PSII excitonic connectivity based on the O–J phase of the OJIP curve [46,47] of paired PSII-LHCIIsc from plants grown in moderate-light supported excitation energy transfer between PSII units (i.e., PSII connectivity) of C_2_S_2_M supercomplexes interacting across the stromal gap [24]. Here, similar analyses on paired supercomplexes isolated from plants grown in high-light and low-light (Figure 5C) further confirmed a higher PSII connectivity in the paired PSII-LHCIIsc with bigger antenna, as shown by the higher difference between W_E_ and W (Figure 5D, Appendix A), and the greater values of the connectivity parameters *C*, *p* and *ω* (Appendix A) of the low-light PSII-LHCIIsc sample (for technical details see Section 4).

## 3. Conclusions

Taken together, our results suggest a diversified strategy adopted by plants to acclimate to different irradiances while maintaining *grana* stacked. In plants grown in low-light, the larger and thicker *grana* membranes accommodate mainly the paired supercomplexes with bigger antenna (C_2_S_2_M_2_)_2_, which have an arrangement that maximizes the exposed area of LHCIIs for light harvesting and increases the PSII connectivity to optimize light-use efficiency. In these plants, the mutual interaction of stromal-exposed N-terminal loops of two facing Lhcb4 subunits in the (C_2_S_2_M_2_)_2_ supercomplex provides a physical anchor in the stromal gap to strengthen the *grana* stacking. In plants grown in high-light, the narrower and thinner *grana* membranes accommodate mostly the paired supercomplexes with smaller antenna (C_2_S_2_)_2_, which have an arrangement that reduces the light-exposed area of LHCIIs, by increasing the overlap of the S-S trimers. This almost complete overlap of LHCII trimers in the (C_2_S_2_)_2_ supercomplex may play a key role in maintaining the *grana* stacking in plants long-term acclimated to high irradiances. In addition, in these plants the partial occupancy of the Lhcb4 position in the (C_2_S_2_)_2_ by the Lhcb4.3 isoform may constitutively reduce the binding affinity for additional M-LHCII trimers thus exerting a photoprotective function for PSII under excessive constant irradiation.

## 4. Materials and Methods

### 4.1. Plant Growth and Isolation of Thylakoid Membranes

*Pisum sativum* L. (pea) plants were grown for 3 weeks inside a growth chamber (MLR-351H SANYO, Osaka, Japan) at 20 °C and 60% humidity under 30 μmol photons m^−2^ s^−1^ (L) and 750 μmol photons m^−2^ s^−1^ (H) of light (8 h/day). The L condition was supplied by turning on three fluorescent lamps (FL40SS W/37, SANYO, Osaka, Japan) in the growth chamber and the H condition by four LEDs (LXR7-SW50, AGON LIGHT, Alessandria, Italy) mounted inside the growth chamber.

All the fully developed leaves were harvested at pre-dawn and immediately used to isolate the thylakoid membranes, according to the following protocol that maintains the *grana*-stack organization. These samples are hereafter referred to as stacked thylakoid membranes. Pea leaves were grinded in a solution made of 50 mM HEPES pH 7.5, 300 mM sucrose and 5 mM MgCl_2_. The suspension was filtered and the filtrate was centrifuged 10 min at 1500× *g*. After one wash by centrifugation in the same buffer, the pellet was homogenized in a solution made of 5 mM MgCl_2_ and diluted 1:1 with 50 mM MES pH 6.0, 400 mM sucrose, 15 mM NaCl and 5 mM MgCl_2_, and finally centrifuged 10 min at 3000× *g*. The resulting pellet of stacked thylakoid membranes was washed by centrifugation in 25 mM MES pH 6.0, 10 mM NaCl and 5 mM MgCl_2_, and finally resuspended in 25 mM MES pH 6.0, 10 mM NaCl, 5 mM MgCl_2_ and 2 M glycine betaine (T buffer) at a Chl concentration above 3 mg mL^−1^. The entire isolation procedure was performed at 4°C under dim green light. Chl concentration was determined by absorption spectroscopy after extraction in 80% (*v*/*v*) acetone [48].

### 4.2. Cryo-ET Sample Preparation, Data Collection and Tomogram Reconstruction

Thylakoid membranes isolated from pea plants grown in high-light and stored at a Chl concentration of ~3 mg mL^−1^ in T buffer, as described above, were diluted 25-fold in a T buffer devoid of glycine betaine to facilitate vitreous ice formation during grid preparation. Then, thylakoids were mixed 1:1 (*v*/*v*) with 10 nm gold particles used as fiducial markers. 3 µL of sample were applied to glow-discharged (1 min O_2_/Ar 11.5/35.0 sccm) Quantifoil R2/1 (Cu 200 Mesh) carbon grids within the chamber of a Vitrobot (mark IV, FEI-Thermo Fisher Scientific, Hillsboro, OR 97124, USA). After 15 s incubation at 100% humidity and 4 °C, excess solution was blotted from both sides for 3 s (blotting force −5), and the grid was plunge-frozen in liquid ethane. Grid vitrification optimization has been performed on a Tecnai G2 F20 (Thermo Fisher Scientific, Hillsboro, OR 97124, USA) equipped with Schottky emitter (maximum acceleration voltage 200 keV), automatic cryo-box and 4 Mpixel UltraScan 1000CCD camera (Gatan, Pleasanton, CA 94588, USA). The grid preparation was performed under dim green light.

Tomograms were collected on a Titan Krios (FEI- Thermo Fisher Scientific, Hillsboro, OR 97124, USA) microscope operated at 300 kV using Tomography 3.0 software (Thermo Fisher Scientific, Hillsboro, OR 97124, USA). The microscope was equipped with a GIF Quantum energy filter, a Gatan K2 Summit direct detection camera (Gatan, Pleasanton, CA 94588, USA) and a Volta phase plate (VPP). Single-axis tilt series were recorded at 3° increment over a range of ±60° in two halves (0° to −60° and 0° to +60°) with a cumulative electron dose of 80 e^−^/Å^2^. Close to-focus VPP data collection was performed on the K2 detector in super-resolution mode for each tilt series, made of 41 tilts fractionated over 10 fractions, at a calibrated magnification of 53,000× *g* corresponding to a magnified pixel size of 1.34 Å. Frames from the K2 detector have been drift corrected and dose weighted with MotionCor2 (MRC Laboratory of Molecular Biology, Cambridge, United kingdom) [49]. Tilt series assembling and tomograms generation have been performed by both WBP and SIRT using IMOD 4.9 (University of Colorado, Boulder CO 80309 USA) [50].

### 4.3. Isolation of PSII-LHCII Supercomplexes

PSII-LHCIIsc were isolated according to a previously optimized protocol [33] with minor modifications. Briefly, stacked thylakoid membranes at a Chl concentration of 1 mg mL^−1^ were solubilized with 50 mM *n*-dodecyl-α-D-maltoside (α-DDM) for 1 min at 4 °C in the dark. The protease inhibitor phenylmethylsulphonylfluoride (500 mM) was added during the solubilization. Solubilized thylakoid membranes were centrifuged 10 min at 21,000× *g* at 4 °C and 700 μL of the supernatant were added to the top of a linear sucrose gradient, prepared by freezing at −80 °C and thawing at 4 °C ultracentrifuge tubes filled with a buffer made of 0.65 M sucrose, 25 mM MES pH 5.7, 10 mM NaCl, 5 mM MgCl_2_ and 0.03% (*w*/*v*) of α-DDM (SG buffer). Thylakoid fractionation was performed by centrifuging the tubes 12 h at 100,000× *g* at 4 °C (Surespin 630 rotor, Thermo Fisher Scientific, Hillsboro, OR 97124, USA). The sucrose band containing PSII-LHCIIsc (i.e., α3, Appendix A) was harvested, concentrated to a Chl concentration above 4 mg mL^−1^ by membrane filtration using Amicon Ultra 100 kDa cutoff device (Millipore, Burlington, MA 01803, USA) and, if necessary, stored at −80 °C. This concentrated sample was directly used, without freezing, for preparation of the cryo-EM grids for the low-light condition. For preparation of the cryo-EM grid for the high-light condition, the sucrose band containing PSII-LHCIIsc was harvested, washed four times by membrane filtration with an Amicon Ultra 100 kDa cutoff device (Millipore, Burlington, MA 01803, USA) with a 25-fold dilution each step with a buffer made of 25 mM MES pH 5.7, 10 mM NaCl, 5 mM MgCl_2_ and 0.03% (*w*/*v*) of α-DDM (EM buffer) to remove the sucrose, concentrated to a Chl concentration above 4 mg mL^−1^ and directly used for cryo-EM.

### 4.4. Cryo-EM Sample Preparation and Data Collection

Concentrated PSII-LHCIIsc were diluted in the EM buffer, the high-light sample to 3 mg mL^−1^ Chl and the low-light sample to 1.5 mg mL^−1^ Chl. 3 μL of sample were applied to glow-discharged Quantifoil R1.2/1.3 (Cu 300 Mesh) carbon grid within the chamber of a mark IV Vitrobot (FEI-Thermo Fisher Scientific, Hillsboro, OR 97124, USA). After 30 s incubation at 100% humidity and 5 °C, excess solution was blotted from both sides for 3 s (blotting force −5), and the grid was plunge-frozen in a liquid ethane/propane mixture. The grid preparation was performed under dim green light.

Data collection for PSII-LHCIIsc in high-light (H dataset) was performed on a Titan Krios (FEI-Thermo Fisher Scientific, Hillsboro, OR 97124, USA) microscope (NeCEN, Leiden, The Netherlands) operated at 300 kV using EPU automated acquisition software (FEI-Thermo Fisher Scientific, Hillsboro, OR 97124, USA). A total of 2734 micrographs were recorded on a Gatan K2 direct electron detector (Gatan, Pleasanton, CA 94588, USA) at 130,000× magnification (image pixel size of 1.1 Å), with a total dose of 40 e^−^/Å^2^ fractionated over 40 frames (5 s exposure, dose rate of 4.2 e^−^/Å^2^/s) and defocus range of −1.2–2.5 μm. Data collection for PSII-LHCIIsc in low-light (L dataset) was performed on a Talos Arctica (FEI- Thermo Fisher Scientific, Hillsboro, OR 97124, USA) microscope operated at 200 kV using EPU automated acquisition software (FEI-Thermo Fisher Scientific, Hillsboro, OR 97124, USA). 1148 micrographs were recorded on a Falcon 3EC direct electron detector (FEI -Thermo Fisher Scientific, Hillsboro, OR 97124, USA) at 57,000× g magnification (image pixel size of 1.82 Å), with a total dose of 50 e^−^/Å^2^ fractionated over 39 frames (1 s exposure, dose rate of 50 e^−^/Å^2^/s), and defocus range of −0.5–2.5 μm.

### 4.5. Cryo-EM Data Processing and 3D Reconstruction

Beam-induced motion was corrected by aligning the image frames with Motioncorr [51] and the aligned images were used for single particle analysis. The contrast transfer function (CTF) was estimated with Gctf (MRC Laboratory of Molecular Biology, Cambridge, United kingdom) [52]. For the H dataset, a total of 80,666 particles were automatically picked from 2734 micrographs using Gautomatch (MRC Laboratory of Molecular Biology, Cambridge, United kingdom). Similarly, for the L dataset, 28,877 particles were selected from 1148 micrographs for further data processing. These particles were classified in 2D and 3D using Relion 2.0 (MRC Laboratory of Molecular Biology, Cambridge, United kingdom) [53]. After 2D classification, 15,950 particles for the H dataset and 1180 particles for the L dataset corresponding to broken and unpaired PSII-LHCIIsc were discarded, resulting in 64,716 paired particles for the H dataset and 27,697 paired particles for the L dataset. The 2D averages assigned to the paired PSII-LHCIIsc were used to generate several initial models using the e2initialmodel.py script of EMAN2 [54] and subsequently select among them that with the highest score to be used as an unbiased low-resolution 3D template for refinement and classification. After 3D classifications in Relion (K = 10), for the H project 27,942 particles were assigned to (C_2_S_2_)_2_-H supercomplexes, 3999 particles to (C_2_S_2_)_2_-H with visible connections spanning the stromal gap and 16,825 particles to (C_2_S_2_M)_2_-H supercomplexes (respectively, class 1, 2 and 3 in Appendix A representing the 43%, 6% and 26% of total particles). Similarly, for the L project 11,094 particles were identified as (C_2_S_2_M_2_)_2_-L supercomplexes, 3546 particles as (C_2_S_2_M)_2_-L supercomplexes and 11,177 particles as (C_2_S_2_M_1-2_)_2_-L supercomplexes made of a combination of C_2_S_2_M_2_ and C_2_S_2_M (respectively, class 1, 2 and 3 in Appendix A representing the 40%, 13% and 40% of total particles). For the H project the (C_2_S_2_)_2_-H particles in class 1 and for the L project the (C_2_S_2_M_2_)_2_-L particles were further processed using the cryoSPARC homogenous refinement algorithm (Structura Biotechnology Inc. Toronto, Canada) [55] and C2 point group symmetry giving final global resolutions, based on the gold-standard Fourier Shell Correlation (FSC = 0.143) criterion [56], of 6.5 Å and 13.1 Å respectively. For the H project, the local reconstruction of one of the two C_2_S_2_ in the map of the (C_2_S_2_)_2_-H supercomplex at 6.5 Å produced the C_2_S_2_-H map with an overall resolution of 3.8 Å (Appendix A). Local resolutions of the density maps were calculated in cryoSPARC [55].

### 4.6. Modelling and Bioinformatics Tools

UCSF Chimera 1.14 (University of California, San Francisco, CA 94110 USA) [57] was used to egment the 3.8 Å (C_2_S_2_)-H cryo-EM map into different zones corresponding to the different proteins that form the supercomplex, using as template the high-resolution structure of the stacked C_2_S_2_M_2_ from *Pisum sativum* (PDB code 5XNL) [22] devoid of the LHCII-M trimers and the Lhcb6 subunits. Protein sequences with differences in amino acids, as detected by top-down mass spectrometry analyses [17] (performed on the same sample preparations used for cryo-EM), with respect to this high-resolution structure, were manually mutated and separately rigid body fitted in the corresponding portion of map. Afterwards, the structure was independently refined into the density map using iterative cycles of Phenix real space refinement [58] and Coot manual adjustment [59]. Finally, all such refined structure was put together into the 3.8 Å density map to undergo the final runs of Phenix real space refinement. Model refinement was performed using symmetry and geometry restraints, enabling secondary structure, rotamer and Ramachandran plot restraints. Geometries of the chlorophylls and haems were strictly restrained towards those present in the pea PSII-LHCIIsc high-resolution structure (PDB code 5XNL) [22], including correct coordination geometry of the metals with protein residues.

### 4.7. Chlorophyll Distance Measurements

The vicinity of Chl molecules with stromal location (Appendix A) was calculated using UCSF Chimera [57] by measuring the Mg-Mg distance between any possible Chls pair within a maximum inter-pigment separation threshold of 90 Å [45]. As a result, some Chls can belong to more than one Chls pair. For the (C_2_S_2_)_2_ supercomplex we used the Chl molecules of our model fitted into the 6.5 Å (C_2_S_2_)_2_-H map, for the (C_2_S_2_M)_2_ and (C_2_S_2_M_2_)_2_ we used two copies of the Chl molecules belonging to the 5XNL PDB structure [22] fitted respectively into the 11 Å (C_2_S_2_M)_2_-H and the 13.1 Å (C_2_S_2_M_2_)_2_-L density maps. To adapt the 5XNL structure to the C_2_S_2_M conformation, the Chl molecules of the LHCII-M trimer (chains 1, 2, 3) and Lhcb6 subunit (chain 4) were removed.

### 4.8. Chlorophyll a Fluorescence Induction Measurements

PSII-LHCIIsc prepared at a Chl concentration of 1.2 µg mL^−1^ in the SG buffer were used for experiments performed according to [24]. Briefly, the fluorescence induction OJIP transient was recorded with a FL3500 double modulation fluorometer (Photon Systems Instruments, Drasov, Czech Republic) illuminating dark-adapted samples for 1 s with continuous actinic light (2400 μmol photons m^−2^ s^−1^, emission peak at 630 nm) at room temperature. The first reliable point of the transient was at t_0_ = 0.02 ms after the onset of illumination. The OJIP curves were analyzed according to methods previously reported [46,47]. These methods rely on the fact that since the shape of the OJIP curve is influenced by excitation energy transfer between PSII units, commonly referred to as PSII connectivity or grouping, the sigmoidicity of its initial phase (i.e., first few microseconds) can be used to provide an estimation of the degree of PSII connectivity [60]. The first 300 μs of the OJIP curves of the PSII-LHCIIsc were used to calculate the normalized O–J phase (labelled W) and the theoretical exponential curve for an unconnected system (labelled W_E_) (Appendix A) [47]. The difference in sigmoidicity from the theoretical unconnected curve W_E_ to the experimental curve W was checked for each sample. The O–J phase of the OJIP curve was also used to estimate the parameters of connectivity *C* (the O–J curvature), *p* (the connectivity among PSII units), and ω (the probability of connectivity between PSII units). For definitions and calculation of these parameters, see Appendix A [46,47]. Results shown are the mean value ± standard deviation of six replicates.

## Figures and Tables

**Figure 1 ijms-21-08643-f001:**
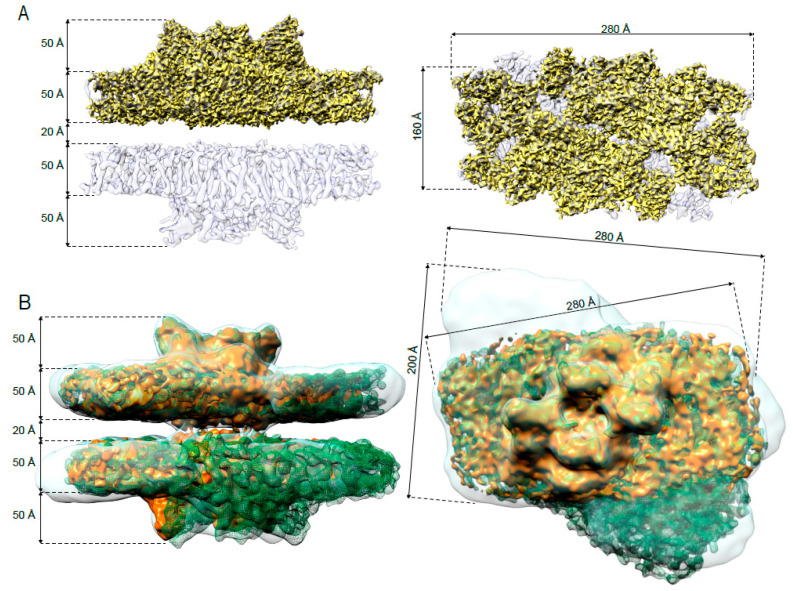
Comparison of the main representative paired PSII-LHCII supercomplexes isolated from plants grown in high-light and low-light. (**A**) Cryo-EM density map of the (C_2_S_2_)_2_-H at 6.5 Å (light violet) with embedded the cryo-EM density map of the C_2_S_2_-H at 3.8 Å (yellow). (**B**) Overlay of the cryo-EM density maps of the (C_2_S_2_)_2_-H at 8.4 Å (orange), (C_2_S_2_M)_2_-H at 11 Å (green mesh) and (C_2_S_2_M_2_)_2_-L at 13.1 Å (transparent light blue). In both panels, the side view (on the left) is along the membrane plane; the top view (on the right) shows the PSII from the lumenal side, normal to the membrane plane.

**Figure 2 ijms-21-08643-f002:**
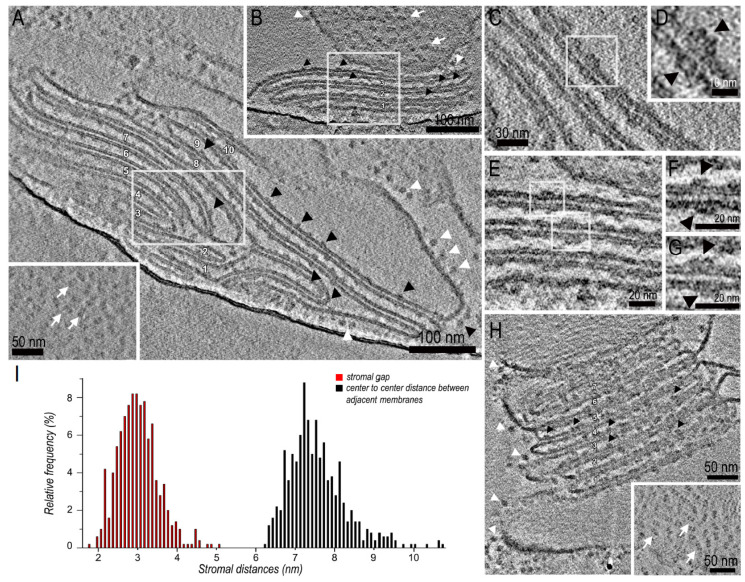
Cryo-electron tomography of isolated stacked thylakoid membranes. (**A**) Average tomographic slice of a stack of 10 thylakoid membranes in side view. The inset shows a tangential averaged tomographic view of a thylakoid membrane. (**B**) Higher magnification of the boxed region in A in a slightly different tomographic section. (**C**) Higher magnification of the boxed region in B showing a paired PSII-LHCII supercomplex in two adjacent thylakoid membranes. (**D**) Average tomographic slice of a stack of 3 thylakoid membranes in side view. (**E**) Higher magnification of the boxed region in (**D**,**F**,**G**) are higher magnifications of the boxed regions in E showing paired PSII-LHCII supercomplexes in two adjacent thylakoid membranes. (**H**) Average tomographic slice of a stack of 7 thylakoid membranes in side view. The inset shows a tangential averaged tomographic view of a thylakoid membrane. (**I**) Distributions of stromal gap (red) and center-to-center distances (black) between adjacent thylakoid membranes across the stromal gap. In the figure, white and black arrowheads point respectively to the ATP-synthase complexes protruding from the *grana* end membranes and to the extrinsic subunits of PSII protruding into the lumen; white arrows point to PSII-LHCII supercomplexes.

**Figure 3 ijms-21-08643-f003:**
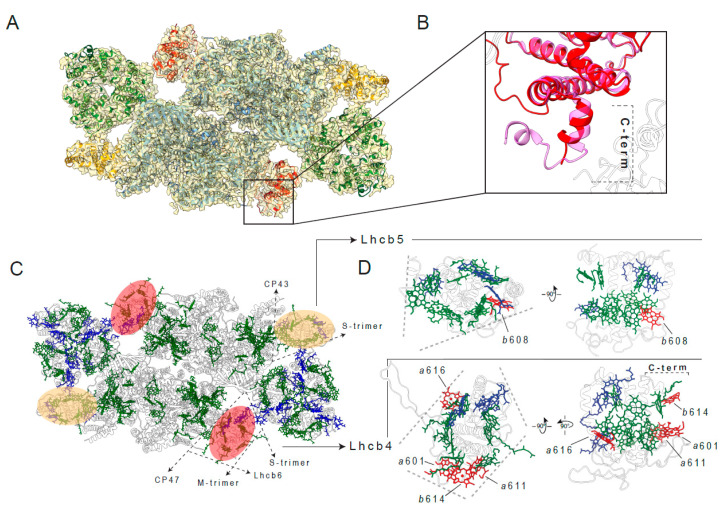
Peculiar features of proteins and chlorophylls of the C_2_S_2_ supercomplex isolated from plants grown in high-light. (**A**) Cryo-EM density map of the C_2_S_2_-H at 3.8 Å viewed from the lumenal side, normal to the membrane plane, and assigned subunits of the PSII dimer in cyan, LHCII-S trimer in green, Lhcb4 in red and Lhcb5 in orange. (**B**) Structural superposition of the C-terminal region of Lhcb4 (red) and Lhcb4 from PDB: 5XNL (chain R, purple) [22]. (**C**) Overview of all chlorophylls *a* (green) and *b* (blue) in the supercomplex shown from the lumenal side. (**D**) Representation of Lhcb4 and Lhcb5 with chlorophyll *a* (green) and chlorophyll *b* (blue) molecules. In panel (**D**), chlorophylls present in the pea C_2_S_2_ supercomplex from PDB: 5XNL (chain R for Lhcb4 and chain S for Lhcb5) [22] and missing in the C_2_S_2_-H are highlighted in red, the top view (on the left) and side view (on the right) of proteins are along the membrane plane.

**Figure 4 ijms-21-08643-f004:**
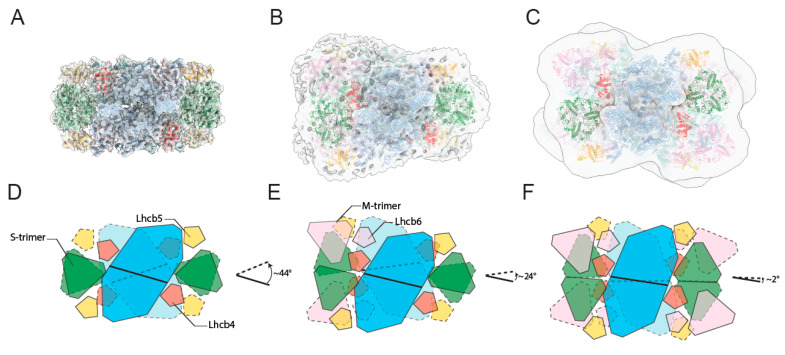
Cryo-EM density maps of paired PSII-LHCII supercomplexes with different antenna size isolated from plants grown in high-light and low-light. Top view towards the lumenal surface of the 3D maps of the supercomplexes (C_2_S_2_)_2_-H at 6.5 Å (**A**), (C_2_S_2_M)_2_-H at 11 Å (**B**), (C_2_S_2_M_2_)_2_-L at 13.1 Å (**C**), fitted with high-resolution structures (PDB: 5XNL devoid of one LHCII-M trimer and one Lhcb6 in panel (**B**); PDB: 5XNL in panel (**C**)) [22], and corresponding schematic representations (**D**–**F**) showing the positions of all fitted supercomplex components and rotational offsets measured between the plane perpendicular to the axis of symmetry of the PSII dimeric core of the lower supercomplex with respect to its counterpart of the upper supercomplex. Subunits coloured as follows: PSII dimer in cyan, LHCII-S trimer in green, Lhcb4 in red, Lhcb5 in orange, LHCII M-trimer and Lhcb6 in pink (dark colours for upper supercomplex, light colours for lower supercomplex). In the scheme, colours match the structures of panels (**A**–**C**); solid lines for upper supercomplex, dashed lines for lower supercomplex.

**Figure 5 ijms-21-08643-f005:**
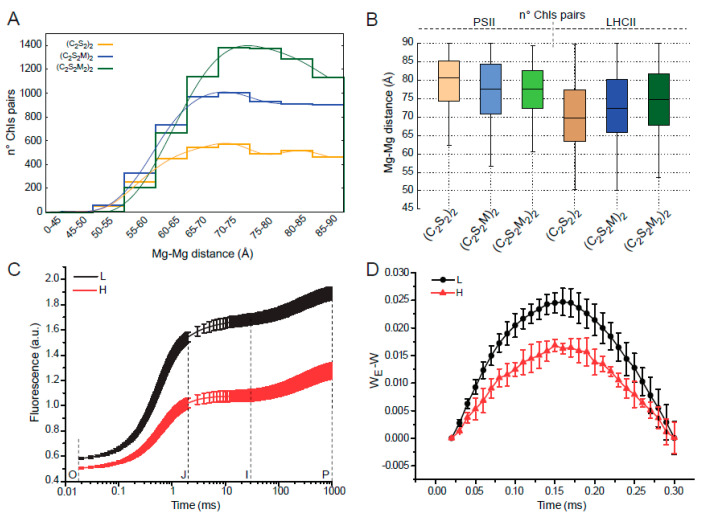
Distance distribution of stromal chlorophylls pairs and PSII excitonic connectivity in paired PSII-LHCII supercomplexes with different antenna size. (**A**) Distribution of Mg-Mg distances calculated for Chls pairs made by a stromal Chl of the upper supercomplex and a stromal Chl of the lower supercomplex within a cut-off of 90 Å and (**B**) corresponding box-plot highlighting the contribution of the Chls pairs, belonging exclusively to PSII cores (left) or LHCII antennae (right), in paired PSII-LHCII supercomplexes of different antenna size. (**C**) Chlorophyll *a* fluorescence induction curves (i.e., the OJIP transient) plotted on a logarithmic time scale for paired PSII-LHCII supercomplexes isolated from H and L samples. (**D**) Estimated PSII energetic connectivity as plot of (W_E_—W), where W and W_E_ (shown in Appendix A) correspond respectively to the normalized O–J phase of the experimental OJIP curve and of the theoretical exponential curve corresponding to the unconnected system (for calculation of W and W_E_, see Appendix A). Graphs in panels C-D display the mean values ± standard deviations of six replicates.

## Data Availability

Cryo-EM maps of the C_2_S_2_-H, (C_2_S_2_)_2_-H at 6.5 Å, (C_2_S_2_)_2_-H at 8.4 Å, (C_2_S_2_M)_2_-H and (C_2_S_2_M_2_)_2_-L were deposited at the Electron Microscopy Data Bank (EMDB) with accession codes EMD-10865, EMD-10866, EMD-10887, EMD-10867 and EMD-10868, respectively. The refined 3D structure of C_2_S_2_-H is available at the Protein Data Bank (PDB) with accession code 6YP7.

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
