# Peer review of "High-Light versus Low-Light: Effects on Paired Photosystem II Supercomplex Structural Rearrangement in Pea Plants"

_ijms, 2020, doi:10.3390/ijms21228643_

Round 1
Reviewer 1 Report
Authors in the revised version improved description of experiments and interpretation of results, especially about the structure of Lhcb4.3 in the PSII C2S2 supercomplex. I am satisfied with their response to my comments as well, and I do not have any further questions.
Reviewer 2 Report
The authors have addressed my comments, making appropriate corresponding changes. I support publication of the manuscript in the current state.
I suggest that the PDB validation report be provided to the editors prior to publication, and I encourage the authors to make unsharpened half-maps publicly available (in addition to the required combined sharpened maps).
This manuscript is a resubmission of an earlier submission. The following is a list of the peer review reports and author responses from that submission.
Round 1
Reviewer 1 Report
The manuscript Grinzato et al. describes structural changes of pairs of photosystem II supercomplexes isolated from pea plants grown under low- and high-light intensities. The authors used the state-of-the-art structural techniques - single particle cryo electron microscopy and cryo electron tomography with very nice data. The manuscript is well written, experiments were carefully performed and analyzed, and several novel information is presented, including the structure of PSII supercomplex with Lhcb4.3 subunits. I have few points, which should be further discussed and clarified in the revised version.
- Single particle electron microscopy revealed novel information about the organization of paired PSII supercomplexes isolated from plants grown under different light intensities. The authors used cryo electron tomography to prove a native origin of the isolated paired PSII supercomplexes. They measured a distance between the two adjacent grana membranes across the stromal space. This distance was similar to the distance between the isolated paired PSII supercomplexes. I have doubts whether this experiment sufficiently supports the native origin of the paired PSII supercomplexes. Is it possible to reveal the paired PSII supercomplexes directly from the tomograms of thylakoid membranes using sub-volume averaging method? This would strongly support the natural occurrence of the paired PSII supercomplexes in the thylakoid membrane.
- The authors used sucrose a gradient ultracentrifugation for isolation of the paired PSII supercomplexes. In general, the amount of PSII supercomplexes is reduced in plants grown under low-light conditions and the amount of PSII supercomplexes remains more-less the same under high-light conditions compared to control plants grown under normal light conditions. Therefore I would expect the lower amount of the paired PSII supercomplexes from the low-light sample, as there we can expect a lower probability to find two PSII supercomplexes close to each other in adjacent thylakoid membranes. Fig. S1 shows a very similar content of these paired PSII supercomplexes in both high-light and low-light plants. Can authors comment on this unexpected finding?
- Single particle electron microscopy revealed the structure of the C2S2 supercomplex from high-light plants, where Lhcb4.3 was structurally detected for the first time. What is the impact of the Lhcb4.3 on the binding of the S and M trimer? How can the authors be sure that Lhcb4.1/4.2 is not present in the data set of the C2S2 supercomplex?
Reviewer 2 Report
Overall:
Grinzato et al. provide cryo-EM and cryo-ET data that provide insight into the interactions of connected Photosystem II supercomplexes between adjacent grana layers in different light conditions. Using EM-based tomograms, they show the frequency of pairing exhibited in different light conditions, and then use maps generated by single-particle analysis of detergent-solubilized paired PSII supercomplexes to suggest the different ways in which pairing might occur to modulate membrane thickness. This is of major interest and I support the publication of this manuscript. I have provided both major and minor comments that I believe should be considered by the editors and authors before publication.
The majority of my major comments revolve around the authors drawing conclusions from the map of Lhcb4.3 that I believe are inappropriate at the local resolution their data exhibits, ~5.0 angstroms where sidechains are not able to be modelled confidently. If the authors are to keep section 2.3, they should make major revisions to the manuscript. Alternatively, they might simply remove much of the content from section 2.3. The manuscript would still maintain high value without it.
Major:
1) Starting at line 268: The authors write, “In our structural model, the Lhcb4.3 isoform can be confidently identified for the presence of a truncated C-terminus with respect to the Lhcb4.1/Lhcb4.2 isoforms, which in Pisum sativum differs for the lack of 14 amino acids and the occurrence of several substitutions (Fig. 3B, Fig. 270 S4), as recently confirmed by top-down mass spectrometry [17].”
Without seeing the density, I am highly doubtful about the confidence of this statement. Though the structure exhibits a 3.8-angstrom global resolution, the local resolution in this region is around 5-angstroms, which is only maybe high enough to observe small peaks for the largest of sidechains (definitely excluding negatively-charged sidechains that do not resolve until much higher resolutions are achieved). The authors do not explicitly mention this resolution limitation but should if they are to draw detailed conclusions from the map.
The authors compare their structure to two other structures. For reference 20, the PSII-LHCII structure from spinach, the global resolution is 3.2-angstroms and the local resolution in the analogous position is around 3.0 angstroms. For reference 22 (where the structure is also shown in Fig. 3 and S4), the global resolution is 2.7-angstroms, and the local resolution in the region of interest is around 4.1-angstroms. Furthermore, in all cryo-EM maps, regions that exhibit high positional variance are always poorly resolved, such as is probably the case with the C-terminal region of Lhcb4.3. In fact, even at much higher resolutions, flexible termini are often unresolved. Considering the very poor local resolution here, and the likely flexibility of the C-terminus, I am not convinced that the model the authors present directly supports their claim that this is Lhcb4.3. To me, the only convincing evidence that supports the authors claim that this is Lhcb4.3 is in their discussion of upregulated transcription in high light (i.e. ref. 16). The authors could make their case more confidently if they show that the larger residues are confidently modelled within their experimental cryo-EM map, perhaps at least couple of Phe residues toward the C-terminus. I am doubtful, though, that this can be achieved at such a resolution.
2) Starting on line 273: The authors write, “The C2S2-H showed a reduced number of chlorophyll molecules with respect to the C2S2 [20] from spinach and the C2S2 module in the C2S2M2 [22] from pea plants grown under unspecified, likely moderate irradiances (Fig. 3C).”
Similar to my comment about the C-terminus, the very low resolution in this region suggests that these Chls are simply not resolved, or even lost during preparation, rather than the author’s suggestion that they are not present. A good example of this even at very high resolution is the 2.44-angstrom resolution structure of PSI acclimated to far-red light by Kato et al. (2020) where they lost PsaJ2 and PsaF2 in sample preparation, and some peripheral Chls are either not resolved or exhibit very low occupancy. If Chl binding is indeed different in Lhcb4.3, it should be relatively obvious in a sequence alignment (Figure S4). However, the sequence comparison suggests that only one Chl, b614, does not maintain the axial coordination conserved in Lhcb4 and Lhcb5. This lack of sequence differences for axial coordination of Chls a601, b608, a611, and a616 is evidence that they are simply not resolved in the cryo-EM map. The authors suggest instead that Chls a601, b608, a611, and a616 are missing as a result of structural changes caused by steric hindrance (b608), and different electrostatic properties of neighboring residues (a601, a611, a616).
b608: Figure S4 panel C argues for the absence of b608, showing the electrostatic potential surface generated from their model. This is a strange choice because the authors do not appear to suggest an electrostatics basis for the absence of a Chl in this site, but rather use the electrostatic surface to suggest that amino acids unique to Lhcb4.3 would occupy this site, causing b608 not to bind. If b608 really is missing, this is absolutely not the way to make such an argument. The surface the authors show is based entirely on showing a radius around their model, and not at all the experimental cryo-EM map. At ~5-angstrom resolution, modelling of sidechains cannot be done very confidently and incorrect modelling would cause the surface to support their conclusion. It is concerning to me that the authors would not show the experimental cryo-EM map. If the authors really want to maintain this argument, they should include a figure showing the experimental cryo-EM map and note the residues in the sequence that cause this steric hindrance.
a601, a611, and a616: Figure S4 panel D argues for the absence of a601, a611, and a616. The authors claim that a difference in nearby electrostatic interactions causes these positions to be vacated, showing these regions on the electrostatic potential surface. They do not, however, describe which residues cause this to occur. Again, the lower resolution probably makes it very difficult to confidently model sidechains, so it is unlikely that features others than secondary structure can even be modelled here, so I doubt the authors have the ability to discuss these interactions. If they do, this should be included, and the cryo-EM map for the residues discussed should be shown to support their confident model placement.
This also makes the rest of the discussion through line 306 erroneous. If the authors cannot make these changes in major comments 1 and 2, I think they should consider removing section 2.3 which I still think makes a nice contribution to the literature.
3) Figure S2: The authors should add more numbers to the local resolution scale so that the reader can see the local resolution in the regions they are referring to. For example, 3.3, 4.1, 4.9, 5.7, 6.5 angstroms.
4) Supplementary Tables 2 and 3 are not provided in the Supplementary Information document I have been provided.
5) A PDB validation report has not been provided.
Minor:
Line 52: Typo, “associates, as a dimeric core…”
There is a typo in Ref. 21.
Line 74: A good supporting reference for the first sentence of this paragraph would be a recent review: “Light harvesting in oxygenic photosynthesis: Structural biology meets spectroscopy” by Croce and van Amerongen (2020).
Figure 2: When the authors write that the black arrowheads correspond to OECs, do they mean the extrinsic subunits of PSII? If so, I suggest changing the language as such.
Figure 2: The white box in panel B should be adjusted to reflect panel E.
Figure 2: The description says that C is the white box in A, magnified. However, this does not look to be the case. Is it only a smaller section and rotated? Maybe the authors could adjust the box in A to reflect C.
Figure 2: Presently, it is confusing that A, C, and D are the same region magnified, and B, E, and F/G are the same region magnified. The authors might consider rearranging the panels so that A, B, and C are progressively increasing magnification of the same region, and D, E, F/G are progressively increasing magnification of the same region.
Figure 2: If black arrowheads are always PSII extrinsic subunits, white arrowheads are always ATP synthase, and white arrows are PSII dimers, the authors might consider only saying that once at the beginning or end of the description instead of stating the same thing multiple times causing the figure description to be verbose.
Line 182: The OEC does not extend into the lumen. The extrinsic subunits of PSII do. As in the Figure 2 legend, I suggest changing the wording.
Line 188: I suggest changing the word “prove” to “exemplify”.
Line 190: I suggest changing the word “accordance” to “agreement”.
Line 198: Typo - “portion” should be “portions”.
Line 199: I suggest replacing the word “local” with “isolated”.
Line 208: I suggest changing the word “accordance” to “agreement”.
Figure 3: In panel A, red and orange are very difficult to differentiate. I suggest adjusting the color.
Line 323: I suggest replacing the word “saying” with “noting”.
Line 345: Typo – “be the primary responsible for” should be “be primarily responsible for”…
References: All of the Science references have “(80-. ).” unintentionally